# Deciphering TAL effectors for 5-methylcytosine and 5-hydroxymethylcytosine recognition

Yuan Zhang[1,2], Lulu Liu[1], Shengjie Guo[1,2,3], Jinghui Song[1], Chenxu Zhu[1], Zongwei Yue[1], Wensheng Wei[1,2,3,4] & Chengqi Yi[1,4,5]

DNA recognition by transcription activator-like effector (TALE) proteins is mediated by tandem repeats that specify nucleotides through repeat-variable diresidues. These repeat-variable diresidues form direct and sequence-specific contacts to DNA bases; hence, TALE–DNA interaction is sensitive to DNA chemical modifications. Here we conduct a thorough investigation, covering all theoretical repeat-variable diresidue combinations, for their recognition capabilities for 5-methylcytosine and 5-hydroxymethylcytosine, two important epigenetic markers in higher eukaryotes. We identify both specific and degenerate repeat-variable diresidues for 5-methylcytosine and 5-hydroxymethylcytosine. Utilizing these novel repeat-variable diresidues, we achieve methylation-dependent gene activation and genome editing in vivo; we also report base-resolution detection of 5hmC in an in vitro assay. Our work deciphers repeat-variable diresidues for 5-methylcytosine and 5-hydroxymethylcytosine, and provides tools for TALE-dependent epigenome recognition.

[1] State Key Laboratory of Protein and Plant Gene Research, School of Life Sciences, Peking University, Beijing 100871, China. [2] Biodynamic Optical Imaging Center, Peking University, Beijing 100871, China. [3] Beijing Advanced Innovation Center for Genomics, Peking University, Beijing 100871, China. [4] Peking-Tsinghua Center for Life Sciences, Peking University, Beijing 100871, China. [5] Synthetic and Functional Biomolecules Center, Department of Chemical Biology, College of Chemistry and Molecular Engineering, Peking University, Beijing 100871, China. Yuan Zhang, Lulu Liu, Shengjie Guo and Jinghui Song contributed equally to this work. Correspondence and requests for materials should be addressed to W.W. (email: wswei@pku.edu.cn) or to C.Y. (email: chengqi.yi@pku.edu.cn)

Transcription activator-like effectors (TALEs) are virulence factors from plant pathogenic bacteria *Xanthomonas* and can reprogram the eukaryotic genome[1, 2]. TALEs contain DNA-binding domains composed of a variable number of tandem repeats[3]. Remarkably, each repeat comprises 33–35 amino acids of consensus sequence, except for the two hypervariable amino acids at position 12 and 13 (repeat-variable diresidues or RVDs)[4, 5]. TALEs bind DNA in a sequence-specific manner and RVDs determine the nucleotide specificity[4, 6]. Exploiting their modular DNA-recognition property, TALEs can be fused with functional domains, such as transcription activators[7, 8], repressors[9, 10] or nucleotide endonucleases[11, 12], to create programmable gene editing tools. Initially, the RVD–DNA recognition code has been partially deciphered using experimental and computational approaches[4, 6]; and it was found that the four most commonly used RVDs NI, NG, HD, and NN preferentially binds to A, T, C, and G/A, respectively[4, 6]. Recently, others and we have performed RVD screening that covers all possible combinations of amino acid diresidues, hence revealing the complete RVD–DNA recognition code[13, 14].

Besides the four canonical nucleotides, mammalian genomes also contain modified DNA bases. For instance, 5-methylcytosine (5mC), which is known as the fifth DNA base, is an important epigenetic marker that regulates gene expression (Fig. 1a)[15, 16]. 5mC can be sequentially oxidized by the ten–eleven translocation (TET) family proteins to produce 5-hydroxymethylcytosine

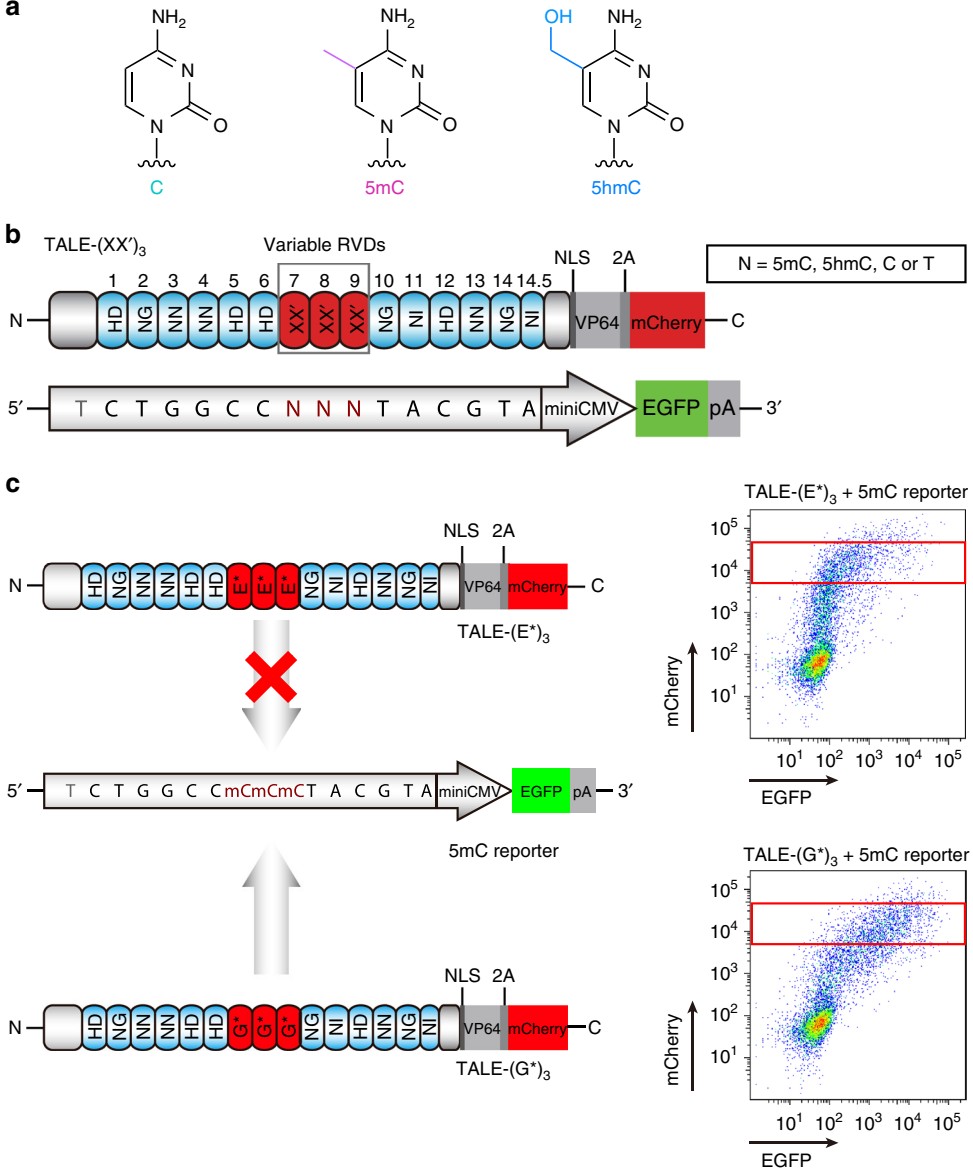

**Fig. 1** The screening platform to completely evaluate all potential TALE RVDs in recognizing modified cytosines. **a** Chemical structures of C, 5mC, and 5hmC. **b** The screening system for novel RVDs against modified cytosine is composed of a TALE activator and a GFP expression reporter. The TALEs contain 14.5 repeats fused with VP64. The 7th to 9th RVDs are substituted by the testing RVD XX′, which makes an archive of 420 TALEs. The X and X′ represent for the 12th and 13th residues, respectively. The 5mC and 5hmC reporters are linear DNA fragments amplified utilizing a forward primer with 5mC or 5hmC embedded on position 7th to 9th and a normal reverse primer. The C and T reporters are circular DNA described as previous[13]. **c** When the customized TALE does not bind to the reporter, for instance, between TALE-(E*)₃ and the 5mC reporter, the GFP expression is at a basal level. In contrast, when the TALE binds tightly to the reporter, as shown for TALE-(G*)₃ to the 5mC reporter, the GFP expression is up-regulated. mCherry intensity implies the quantity of transfected TALE-(XX′)₃ plasmids

(5hmC), 5-formylcytosine (5fC) and 5-carboxylcytosine (5caC), the latter two are substrates of thymine DNA glycosylase and are eventually restored to unmodified cytosine[17–22]. 5hmC composes ~ 1%-10% of modified cytosines and is believed to be a stable epigenetic mark; dysregulation of 5hmC are frequently observed in cancer[23].

TALE proteins have been shown to recognize modified DNA bases[24–26]. For instance, NG or N* (the asterisk represents the deletion of the 13th amino acid) have been reported to tolerate 5mC in the cognate DNA target[25, 27–31]; the combination of NG/N* and HD was used to discriminate 5mC/5hmC from C in an in vitro assay[32]. A recent study also reports that TALE proteins with size-reduced repeat loops (G*, S*, and T*) could bind to C, 5mC, 5hmC, 5fC, and 5caC with similar affinities[33, 34]. In the crystal structures of TALE–DNA complex, the RVD loops contact the duplex major groove, in which the first residue stabilizes the proper loop conformation and the second residue makes a direct base-specific contact[35, 36]. These observations illustrate the potential of TALEs in discriminating canonical and modified cytosine bases. However, the full potential of RVDs in recognizing 5mC and 5hmC has not been explored.

## Results

**Screening platform to evaluate RVDs for modified cytosines**. To allow unbiased assessment of RVDs for 5mC and 5hmC recognition, we utilized the screening platform that we developed previously[13]. This platform is comprised of an archive of 400 TALE-VP64-mCherry constructs with three tandem RVDs from the 7th to the 9th repeat, designated as TALE-(XX')3. Because of previous findings that N* can recognize 5mC, we additionally assembled 20 TALE-(X*)3, in which the 13th residue is absent. Thus, we expanded the TALE archive to 420 RVD combinations. Another challenge for the identification of methylcytosine-accommodate RVDs is to prepare the chemically modified DNA reporter constructs. Because reporter plasmids amplified in *Escherichia coli* do not possess the desired DNA modifications, we chemically synthesized 5mC and 5hmC containing primers and generated the reporters by PCR. Hence, the linear reporters are comprised of TALE-binding sites with site-specifically modified 5mC or 5hmC, miniCMV promoter, EGFP coding sequence and polyA signaling (Fig. 1b; Supplementary Fig. 1 and Supplementary Methods).

To measure the binding affinities of the 420 RVDs to 5mC and 5hmC, we introduced each of the 420 TALE-(XX')3 constructs together with one of the three EGFP reporter DNA (containing either C, 5mC or 5hmC) into HEK293T cells. Both the EGFP and mCherry fluorescence levels were measured using FACS analysis (Fig. 1c). By comparing the fold change of EGFP expression of each RVD to the corresponding base with the basal level, we determined the specificities of the 420 RVDs in TALE-(XX')3 constructs to C, 5mC and 5hmC, respectively. A total of 1260 data points for C, 5mC and 5hmC were summarized in a heat map, along with the 420 data points of T from our previous work[13] (Fig. 2a; Supplementary Fig. 2). To our knowledge, this is the first comprehensive screening for TALE RVDs in recognizing 5mC and 5hmC.

**Specific and degenerate RVDs for 5mC and 5hmC recognition**. We identified multiple 5mC binders that showed high activation efficiency in the above in cellulo assay and group them into three categories based on the amino acid residue at the 13th position: Gly-containing RVDs (NG, KG, and RG), Ala-containing RVDs (HA and NA) and deletion-containing RVDs (N*, K*, H*, R*, Y*, and G*). Within the Gly- and deletion-containing RVDs, there are both universal (recognizing 5mC, 5hmC and unmodified C)

and degenerate (recognizing both 5mC and 5hmC) RVDs; interestingly, the two Ala-containing RVDs (HA and NA) are selective for 5mC. Previous studies used NG (the natural binder of T) and N* for 5mC recognition; while we also identified the two RVDs in our screening, their binding affinities for 5mC are outcompeted by many RVDs reported in this study. For instance, HA, NA, and X* (X denotes K, H, Y, and G) all demonstrate higher activation efficiencies for 5mC-containing reporters. We did find that RVDs in the three categories also bind to unmodified T; this is not surprising giving that they have either amino acid residues with small side chains or deletion of this residue at the 13th position.

No RVD has previously been reported to selectively bind 5hmC. As we mentioned above, we identified both degenerate and universal RVGs that bind well to 5hmC. Among them, up to ~ 15 fold induction was observed for these 5hmC binders, demonstrating strong affinity towards 5hmC. In addition, we also observed a new group of 5hmC-binding RVDs with serine at the 13th residue (FS, YS, and WS). Although their abilities in activating 5hmC-containing reporters are lower than the universal binders, they preferentially recognize 5hmC than 5mC, offering the potential of positive and selective 5hmC recognition. Taken together, it appears to us that the universal and degenerate binders for 5mC and 5hmC prefer to contain a glycine or a deletion at the 13th position, while the specific binders for 5mC and 5hmC have an alanine or serine residue at their 13th position.

**Affinities and specificities of RVDs for modified cytosines**. To further validate DNA recognition by the identified RVDs, we carried out an in vitro protection assay (Fig. 3a). In this experiment, we chemically synthesized DNA oligos that contain either C, 5mC or 5hmC at a defined position (within a restriction site); an endonuclease was added to the DNA probes in the presence of varying concentrations of TALE proteins. Binding of TALE proteins to their cognate cytosine bases will inhibit DNA cleavage by the endonuclease, thereby resulting in a protected full-length band and a cleaved-DNA band during denaturing PAGE analysis. The protection efficiency, in the form of inhibition constant ($K_i$, which is the inverse measurement for binding affinity), was then calculated for each RVD.

We first optimized the assay using HD, the high-affinity natural binder for unmodified cytosine. A low $K_i$ was observed for C, while the $K_i$ for 5mC and 5hmC were at least 30-fold greater (Fig. 1b; Supplementary Fig. 3c), demonstrating the capability of the protection assay in quantitative assessment of the binding affinity. We also showed in this in vitro assay that NG and N* can only bind to 5mC but not 5hmC. We then selected representative RVDs from our screening for further evaluation. The 5mC-specific RVD HA shows the lowest $K_i$ for 5mC and also demonstrates ~ 5–7-fold of selectivity against C and 5hmC in this in vitro assay. The 5hmC-specific RVD FS shows ~ 5–6-fold of selectivity against C and 5mC, although its binding affinity to 5hmC appears to be less strong than HA to 5mC. In addition, the degenerate RVD RG exhibits comparable protection for 5mC and 5hmC, while the universal R*, which binds to C, 5mC and 5hmC, demonstrates similar affinity for all. To further examine the binding affinities of TALE RVDs to different cytosines, we performed electrophoretic mobility shift assay (EMSA). The results again agree well with our in cellulo observations (Supplementary Fig. 3d, e).

**Gene activation via RVDs in a methylation-dependent manner**. To explore the potential of these RVDs in recognizing cytosine methylation in vivo, we investigated their performance in targeted

gene activation in human cells. A previously developed TALE-VP64 design was used to achieve specific gene activation[37]. Utilizing the existing methylation data from the USCS database, we selected the *TET1* gene, whose promoter has a high methylation level in HeLa cells but is hypomethylated in HEK293T cells (Fig. 4a). In HeLa cells, the 5mC-specific HA, the degenerate RG and the universal R* all significantly activated *TET1* expression, with RG achieving about 10-fold activation (Fig. 4b). All the three RVDs identified in this study demonstrated better performance when compared to NG and N*. In addition, HD did not significantly upregulate *TET1* expression. In HEK293T cells, HD bound well to the hypomethylated *TET1* promoter and further enhanced its expression (despite of its already high expression

level). Consistent with our expectation, HA and RG did not affect gene expression, while the universal R*, whose affinity to unmodified C is lower than HD, mildly upregulated gene expression. Because NG and N* can poorly discriminate unmodified C, they also slightly activated *TET1* expression.

To further explore the utility of these RVDs in activating targets with medium level methylation, we then constructed TALE-activators targeting the promoter region of the *LRP2* gene, which is medium methylated in HeLa cells and is again hypomethylated in HEK293T cells (Fig. 4c). In addition, this region contains only two CpGs and hence is more challenging for RVD-mediated discrimination. We again observed significant gene activations in HeLa cells for the 5mC-binding RVDs. In

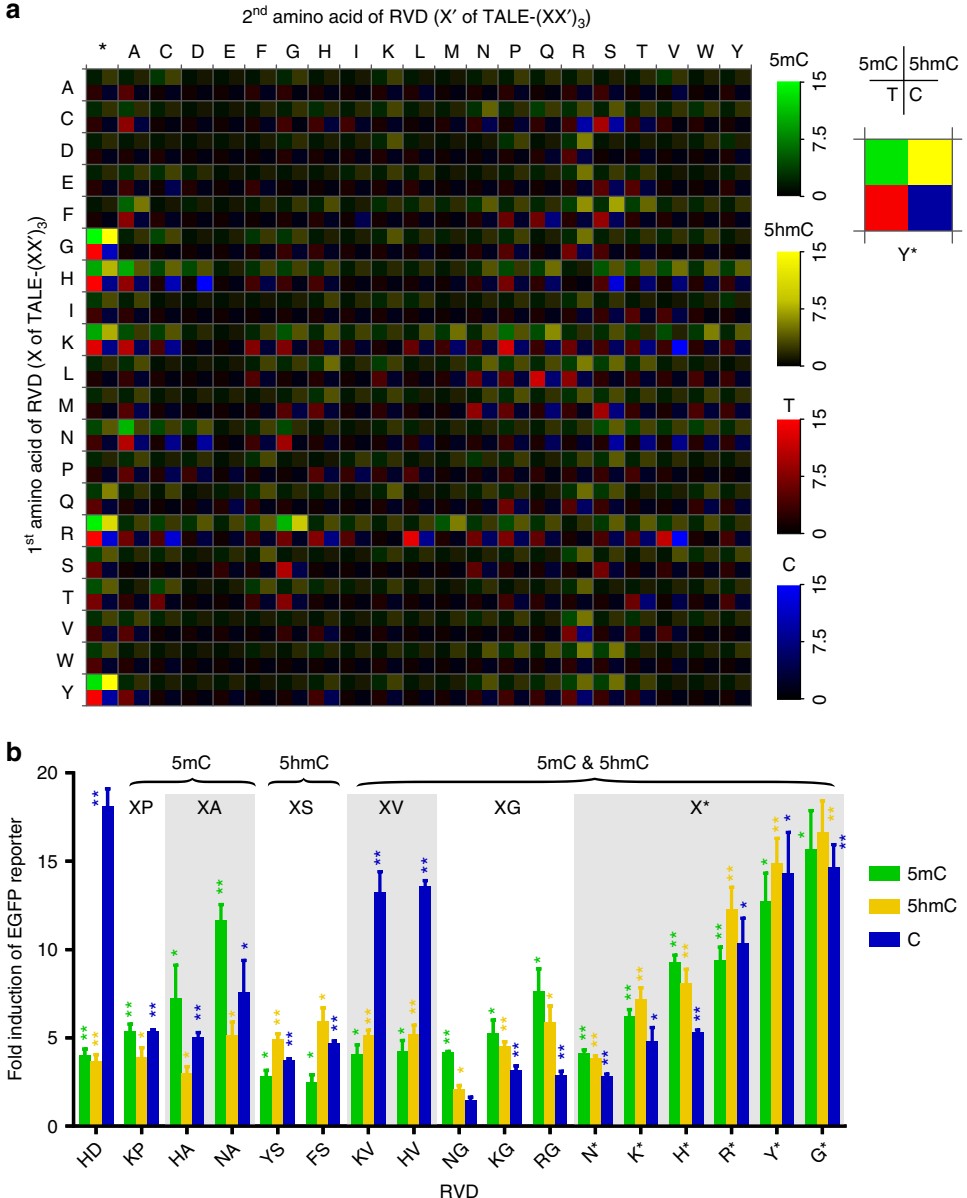

**Fig. 2** Complete assessment of the efficiencies and specificities of TALE RVDs towards 5mC and 5hmC. **a** A heat map summarizing the screening data of 5mC and 5hmC. For easy comparison, the results for unmodified C and T reporters are also shown. EGFP activities from different reporters are coded using different colors representing the reporter identities (5mC, *green*; 5hmC, *yellow*; T, *red*; C, *blue*), and the brightness of the colors indicates the fold induction of reporters by TALE-(XX')₃ normalized to the basal levels. The single-letter abbreviations for the amino acids are used. The *upper right side* legend shows the specific heat map of Y*. Starting from the *top left*, the DNA reporter is 5mC, 5hmC, T and C, respectively. **b** Modified cytosine preference of RVDs. RVDs are clustered according to base preference and are categorized by the 13th residue in each cluster. The data are means + s.d., n = 3; *P < 0.05, and **P < 0.005 two-tailed unpaired Student's *t* test

HEK293T cells, only HD and the universal RVD R*, but not the 5mC-binding RVDs, activated the expression of the *LRP2* gene. Hence, the identified RVDs are capable of distinguishing medium methylated sites from unmethylated sites in vivo.

**Methylation-dependent genome editing using the novel RVDs**. To examine the possibility of methylation-dependent genome editing, we applied these RVDs in TALEN constructs targeting the human *PLXNB2* gene (Fig. 4e). We chose the second exon of *PLXNB2* that is highly methylated in HeLa cells (data from UCSC) and evaluated TALEN-mediated DNA cleavage using the indel rates. In accordance with our expectation, the TALEN-HD showed negligible editing efficiency (Fig. 4f), suggesting that the presence of three 5mC modifications within this region efficiently blocked its binding. When the three HD-containing RVDs were replaced by 5mC-binding RVDs (HA, R*, NG and N* were tested), high indel rates were observed (Fig. 4f; Supplementary Fig. 4c). These results suggested that combining the knowledge of the methylation data, these RVDs can be enabled to achieve methylation-dependent genome editing in human cells.

**Locus-specific 5hmC detection using RVDs**. The methylation ratio of cytosine can be qualified by bisulfite sequencing; however, traditional bisulfite sequencing cannot distinguish 5hmC from 5mC[38]. Indirect 5hmC detection using C- and 5mC-binding TALE proteins have been reported previously[32]; to explore the possibility of direct 5hmC detection using TALE proteins that contain 5hmC-recognizing RVDs, we first synthesized model DNA sequences with site-specifically incorporated 5hmC, 5mC and C, and tested the selectivity of 5hmC detection by FS. In an in vitro protection assay, the protected full-length DNA increased linearly as the ratio of 5hmC increased (Supplementary Fig. 5). In contrast, when the ratio of 5mC and C was varied, the protection ratio showed very modest change. These observations suggest that the 5hmC-specific RVD FS may be suitable to detect 5hmC modification in the complex modification situations (the simultaneous existence of at least C, 5mC, and 5hmC for the nucleotide of interest) in genomic DNA samples.

We then utilized FS-containing TALE proteins to achieve locus-specific 5hmC detection in genomic DNA. Considering the complexity of the genomic DNA, we used the CRISPR-cas9 system instead of restriction enzymes to generate DNA cleavage in this protection assay (Fig. 5a). We chose a 10 bp sequence in the intron of mouse *Slc9a9* gene, in which the first cytosine was reported to be highly hydroxymethylated in mES cells[39]. Indeed, the protection efficiency of TALE-FS was much higher than that of TALE-HD (Fig. 5b), indicating that TALE-FS is capable of detecting one single 5hmC site in the complex environment of genomic DNA. To further explore the ability of this approach in

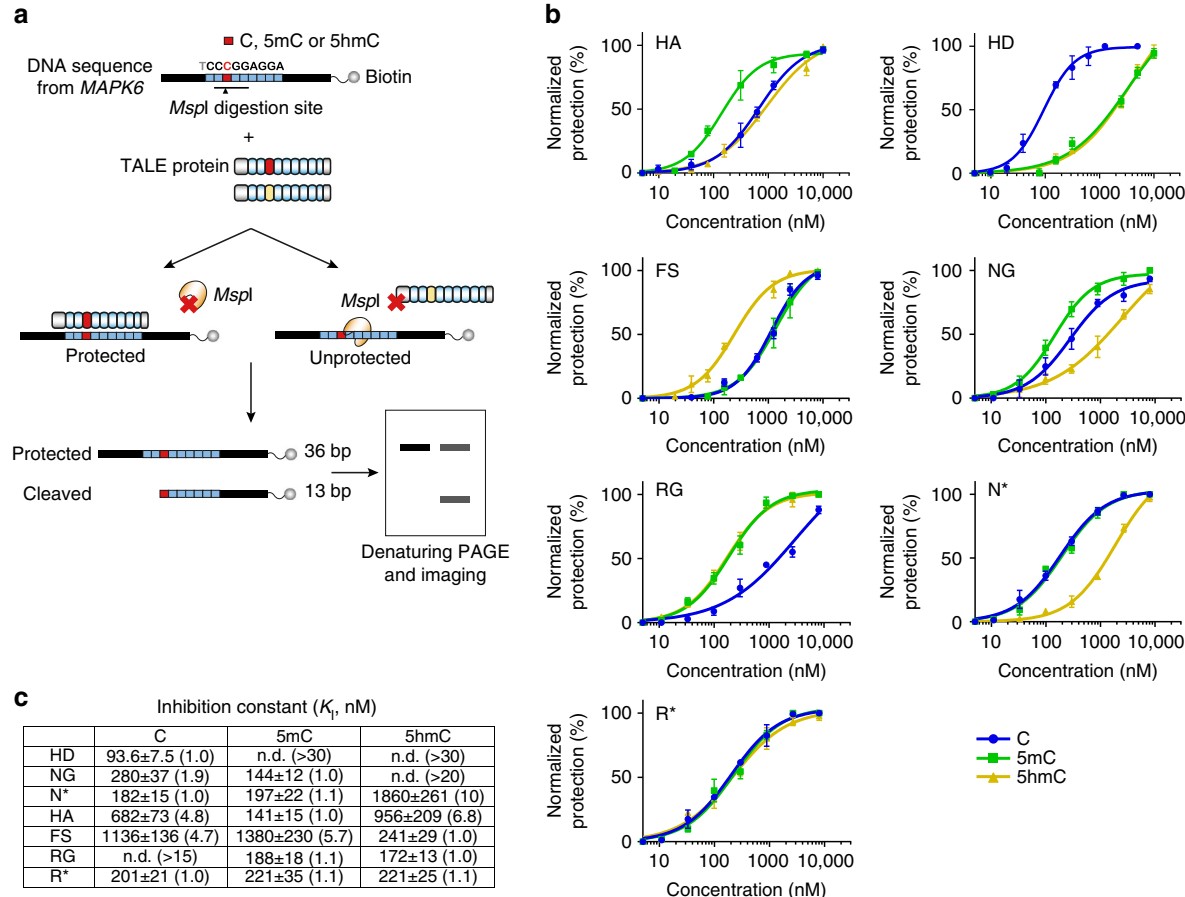

| Inhibition constant ($K_I$, nM) | | | |
|---|---|---|---|
| | C | 5mC | 5hmC |
| HD | 93.6±7.5 (1.0) | n.d. (>30) | n.d. (>30) |
| NG | 280±37 (1.9) | 144±12 (1.0) | n.d. (>20) |
| N* | 182±15 (1.0) | 197±22 (1.1) | 1860±261 (10) |
| HA | 682±73 (4.8) | 141±15 (1.0) | 956±209 (6.8) |
| FS | 1136±136 (4.7) | 1380±230 (5.7) | 241±29 (1.0) |
| RG | n.d. (>15) | 188±18 (1.1) | 172±13 (1.0) |
| R* | 201±21 (1.0) | 221±35 (1.1) | 221±25 (1.1) |

**Fig. 3** Quantitative measurement of DNA recognition by TALE RVDs using an in vitro protection assay. **a** Principle of in vitro protection assay. Briefly, binding of TALE proteins to their cognate cytosine bases will inhibit the endonuclease cleavage, thereby resulting in a protected full-length band and a cleaved-DNA band during denaturing PAGE analysis. **b** The normalized protection efficiency, as measured by the fraction of the uncleaved or protected DNA, are fitted into protection curves of different TALE RVDs. The curves are fitted into specific binding curve with Hill slope (GraphPad). **c** Inhibition constant calculated from **b**. The ratio of each constant to the lowest constant of the same RVD is indicated within the parentheses. The data are means ± s.d., $n = 3$; *$P < 0.05$, and **$P < 0.005$ two-tailed unpaired Student's $t$ test

5hmC detection, we applied this method to the genomic DNAs of additional cell lines whose hydroxymethylation level at the same site was unknown. Comparing to the mESC samples, protection of genome DNAs from these cells was much smaller when relatively low concentration of TALE proteins is present (Fig. 5c), suggesting a lower level of 5hmC for this particular site in these cells. Although the optimized conditions for this assay may be context-dependent, our results showed that TALE proteins containing the identified RVDs can be used to detect the hydroxymethylation status at base-resolution in genomic DNAs.

## Discussion

Despite the wide use of TALE-based technologies, its DNA recognition and specificity is still not fully understood. We showed in this study that DNA binding by TALE proteins is affected by DNA base modifications. Utilizing 420 combinations of RVDs, we identified novel RVDs of unique specificity to 5mC and 5hmC, two important epigenetic markers in higher eukaryotes. The methyl and hydroxymethyl groups do not interfere with base-pairing; yet, their presence in the major groove of DNA duplex impact their interaction with TALE proteins. In other words, the physical interactions between TALE proteins and DNA base pairs offers a unique opportunity to directly "read-out"

the modification status of DNA duplex. As a comparison, another genome editing tool, CRISPR-Cas9, achieves target recognition via base pairing between a single guide RNA (sgRNA) and the DNA target[40], and hence is incapable of detecting the methylation status of the genome. It remains to be determined that besides 5mC and 5hmC, whether TALE RVDs are present for the specific recognition of several other epigenetic DNA modifications in the mammalian genome, including 5-formylcytosine, 5-carboxylcytosine and N6-methyladenosine.

Previous structures of TALE–DNA complex showed that the amino acid at the 13th position is the only residue that directly interacts with the DNA base of the sense strand, whereas the 12th residue serves to stabilize the proper loop conformation during base pair recognition[35, 36]. We showed that, in general, small amino acids (Gly and Ala) or deletion at the 13th position could increase the binding affinity for 5mC. This observation is consistent with previous findings that N* and NG could bind to 5mC. NG is the natural binder of T; some of the specific (FS) and degenerate (RG) RVDs we identified, which has small amino acids at the 13th position, could also accommodate T (Supplementary Fig. 3d, e). Hence, cautions should be taken that DNA sequences with identical flanking sequence but containing either modified cytosines or T could both be targeted during potential

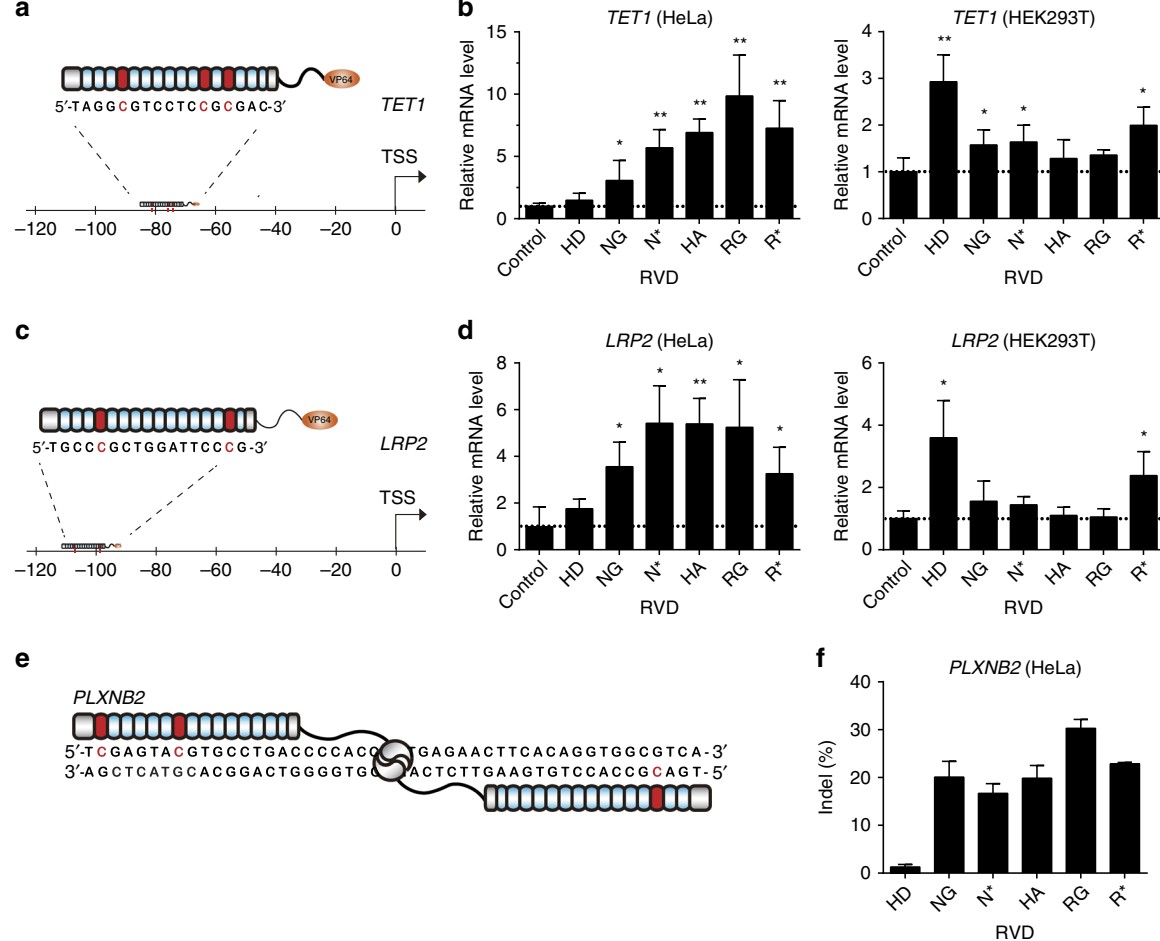

**Fig. 4** Methylation-dependent gene expression activation and genome editing. **a** TALE_TET1 targets a 16 bp DNA sequence ~80 bp upstream of the transcription start site (*TSS*) of the *TET1* gene. All three CpGs (whose Cs are indicated in *red*) in the region are highly methylated in HeLa cells but unmethylated in HEK293T cells. **b** The relative mRNA level of *TET1* in HeLa and HEK293T cells transfected with TALE_TET1 containing different RVDs. **c** TALE_LRP2 targets a 16 bp sequence ~ 100 bp upstream the TSS of the *LRP2* gene. Both the two CpGs in this region contain medium-level methylation in HeLa cells while are unmethylated in HEK293T cells. **d** The relative mRNA level of *LRP2* in HeLa and HEK293T cells transfected with TALE_LRP2 containing different RVDs. **e** The position of TALEN targeted sequence. The methylated CpGs are indicated in *red*. **f** The genome editing efficiency of TALEN using different RVDs. The data are means + s.d., $n = 3$; *$P < 0.05$, and **$P < 0.005$ two-tailed unpaired Student's *t* test

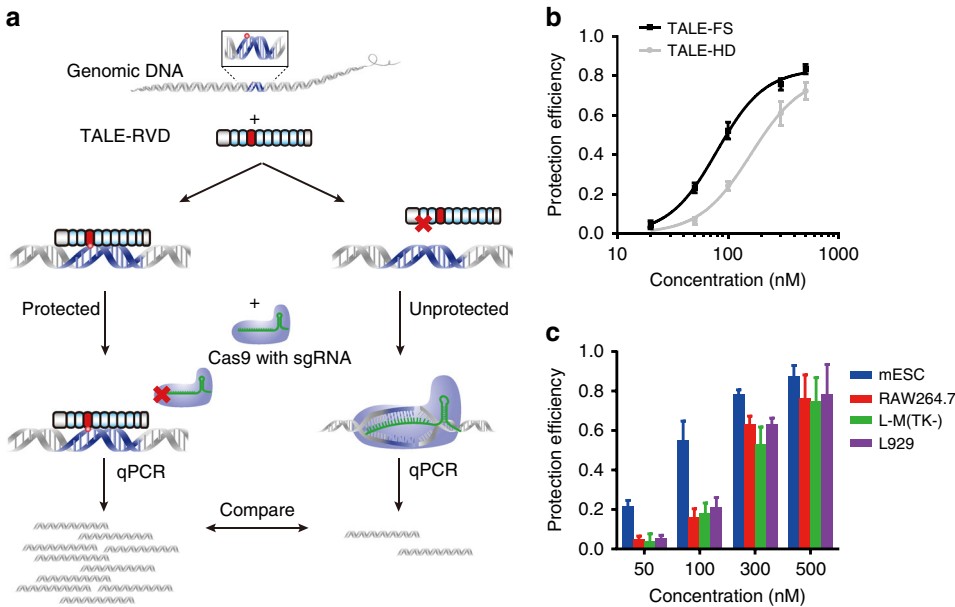

**Fig. 5** Detection of 5hmC in genomic DNA at single-base resolution. **a** Workflow of base-resolution 5hmC detection by novel RVDs. Briefly, the targeted genomic region is protected by TALE, against Cas9-mediated DNA cleavage. **b** The protection efficiency of TALE-FS (*black*) and TALE-HD (*gray*) targeting a single 5hmC site of mESC genome. **c** Protection efficiency of TALE-FS for a single 5hmC site in the genomic DNA of mESC, RAW264.7, L-M(TK−) and L929 cells. At this given site, mESC genome contains the highest 5hmC modification level among the cell lines. The data are means ± s.d., $n = 3$; *$P < 0.05$, and **$P < 0.005$ two-tailed unpaired Student's $t$ test

in vivo applications. It is likely that the absence of bulky side chains at the 13th position could create enough space to accommodate the methyl group of 5mC. However, there are exceptions to this general trend. For instance, we observed very weak 5mC binding affinity for HG, which contains a smaller residue at the 13th position as compared to HD, the natural binder of C. Interestingly, when the His at the 12th position is replaced by Arg (hence, becoming RG), we observed strong binding to 5mC. As a matter of fact, RG also recognizes 5hmC. These observations suggest a potentially more complicated mode of modification recognition by the diresidues. Crystal structures of these RVDs in complex with the modified cytosines are needed to fully understand the mechanism of modification recognition by TALEs.

We demonstrated TALE-mediated, methylation-dependent gene activation and genome editing for several hypermethylated genomic regions. As an important control, when the same regions are devoid of cytosine methylation (in different cells), little gene activation could be observed. Hence, the novel RVDs reported in this study may provide a possibility to manipulate target genes based on their modification status in vivo. It is known that many differentially methylated regions are present and they are involved in many important biological events including genomic imprinting and diseases. Hence, the unique ability of TALE proteins to read out the epigenetic markers might enable future epigenome-dependent applications of TALE in vivo.

## Methods

**Artificial system for RVD screening**. The artificial screening system was composed of three reporters and a TALE-VP64 expression library in which the RVDs of three consecutive monomers in the middle of an artificial TALE array were encoded by the same six randomly synthesized nucleotides (TALE-(XX′)₃). The TALE-VP64 expression library was generated as described before[13]. Reporters are PCR amplified linear dsDNA using chemical synthesized primers consisted of TALE-(XX′)₃-binding sites 5′-CTGGCCNNNTACGTA-3′, in which N represents 5mC or 5hmC, was located immediately upstream of a minimal CMV promoter (PminCMV) and its downstream EGFP gene.

**Protection assay**. For MspI protection assay: each 10 µL reaction contains 1 nM labeled DNA, 1 µL 10× CutSmart Buffer (NEB) and 100 nM NaCl. TALE proteins was added to a final concentration between 10 nM and 8 µM. The binding system was incubated at 25 °C for 30 min. 0.4 U of MspI was then added and the incubation was continued for 15 min. The reaction was quenched by add of 10 µL Formamid followed by heating at 95 °C for 5 min. Protected and cleaved DNA were separated by Urea-PAGE and imaged by Chemiluminescent Nucleic Acid Detection Module Kit (Thermo).

For Cas9 protection assay: each 10 µL reaction contains 50 ng genomic DNA, 1 µL 10× Cas9 nuclease reaction buffer (NEB) and 1 nM DTT. TALE proteins was added to a final concentration between 20 and 500 nM. The binding reaction was incubated at 25 °C for 30 min. A total of 5 µL preincubated Cas9 and sgRNA was added and the incubation was continued at 37 °C for 1 h. The reaction was quenched by heating at 95 °C for 5 min. DNA was purified by Ampure Beads and qPCR analyzed using SYBR Green 2× premix II(Takara) on LightCycler® 96 (Roche).

**Assessment of TALENs-mediated indels**. HeLa cells were seeded in 6-well plates and grown to 60% confluence. For each well, a pair of TALEN plasmids and pmaxGFP (Lonza Group Ltd.) were co-transfected at a ratio of 9:9:2 (0.9: 0.9: 0.2 µg) using Xtreme Gene HP (Roche). The transfected cells were cultured for 3 days before flow cytometric sorting for GFP-positive cells.

TALENs-targeting regions were PCR-amplified from the genome DNA of the isolated GFP positive cells. TALENs-mediated indels were analyzed by T7E1 assay mediated by mismatch-sensitive T7 endonuclease I according to the manufacturer's instructions (New England Biolabs)[41].

**Assessment of TALE-activator-mediated gene activation**. HEK293T and HeLa cells were seeded in 6-well plates and grown to 60% confluence. For each well, 2 µg TALE-activator plasmid was transfected using Lipofactamine® 2000 (Invitrogen). The transfected cells were cultured for 3 days before flow cytometric sorting for mCherry-positive cells.

Total RNA was isolated from mCherry-positive cells and reverse transcribed. Real-time PCR analysis was performed on the ViiATM7 Real-Time PCR System (Applied Biosystems) at standard reaction condition using SYBR Green 2× premix II (Takara).

**DNA synthesis and purification**. Oligo DNA primers were synthesized on an Expedite 8909 DNA/RNA synthesizer using standard reagents including 5mC and 5hmC phosphoramidites (Glen Research). Oligo DNA was deprotected by standard methods recommended by Glen Research Corp. and purified by Glen-Pak DNA purification cartridge.

Synthetic modified oligonucleotides sequences:
F: 5′-G*C*C*AGATATACGCGTTACTGGAGCCATCTGGCCNNNTACGTA GGCGTGTAC-3′, R: 5′-A*G*C*GT CTCCCGTAAAGCACTAAATCGGAACCC TAAAGGGAGC-3′.

The three 5′ phosphodiester bonds of F and R primer are phosphorothioate modified (indicated by *).

Synthesized DNA were validated by high-performance liquid chromatograph (HPLC), briefly: DNA was digested into nucleosides with nuclease P1 (Sigma, N8630) and alkaline phosphatase (Sigma, P4252). The nucleosides were separated on SB-Aq C18 column (Agilent) using 5% to 50% Acetonitrile in 30 min.

**Transfection and flow cytometric analysis.** HEK293T cells (from Stanley Cohen lab at Stanford University) were cultured in DMEM with 10% FBS and 1% penicillin-streptomycin at 37 °C and 5% $CO_2$. Cells were seeded 24 h prior to transfection in 24-well plates at a density of $7 \times 10^4$ cells per well. The cells in each well were co-transfected with 0.15 µg TALE-(XX′)$_3$ plasmid and 0.45 µg reporter DNA using polyethylenimine (PEI). At 48 h post-transfection, cells were collected and analyzed on BD LSR Fortessa flow cytometer (BD Biosciences). Lasers with wavelengths of 488 and 561 nm were used to quantify EGFP and mCherry protein expression, respectively. At least 10,000 events were collected from each sample to obtain sufficient data for analysis. Cells with mCherry fluorescence intensity of $5 \times 10^3$–$5 \times 10^4$ were gated for analysis. mES cells are provided by Fuchou Tang lab at Peking University.

**Construction of TALE-activators and TALENs.** TALE-activators and TALENs with canonical RVDs (i.e., NI, NG, HD and NN) were constructed into pLenti-Lox3.7-TALE and pGL3-TALEN expression vector[37], using the advanced ULti-MATE system[42]. For TALE repeats using novel RVDs, TALE monomers containing novel RVDs were individually synthesized. The final assembly of these TALEs constructs were conducted using the same protocol as above.

**Construction of TALE expression plasmids.** TALE repeats were constructed into TALEN backbone. And fragments containing the N- and C-terminal sequences of TALE with internal repeats were amplified from correspondence TALEN plasmid and cloned into NheI and HindIII sites of pET-28a( + ).

**Protein purification.** The sequence of TALE with varied RVDs was cloned into pET-28a( + ) vector (Novagen). Overexpression of TALE proteins was induced in E. coli BL21 (DE3) by 1.0 mM isopropyl β-D-thiogalactoside (IPTG) when the cell density reached an OD600 of 0.8. After growth at 20 °C for 16 h, the cells were harvested, re-suspended in the buffer containing 25 mM Tris-HCl pH 8.0, and 150 mM NaCl, and disrupted using sonication. The recombinant proteins were purified sequentially through $Ni^{2+}$ -nitrilotriacetate affinity resin (Ni-NTA, GE healthcare) (BufferA: 10 mM Tris-HCl pH 8.0, 300 mM NaCl and BufferB: 10 mM Tris-HCl pH 8.0, 300 mM NaCl and 500 mM imidazole) and HiLoad superdax PG200 (GE Healthcare) (Buffer GF: 10 mM Tris-HCl, pH 8.0, 100 mM NaCl).

**Electrophoretic mobility shift assay.** In order to maintain consistency with our protection assay in Fig. 3, we used the same set of probes, which are derived from the MAPK6 gene and contain one modified cytosine in the middle of the sequence. We supposed the binding affinity can be influenced by the increased number of modified cytosines. We chemically synthesized DNA probes which contain three 5mC, 5hmC, C, and T at defined positions as EMSA nucleic acid targets. We also constructed and purified an additional series of new TALE proteins with varying RVDs to recognize all the three cytosines and T. For EMSA experiments, a 2-fold and then serial 1.4-fold dilution of 5 µM newly constructed TALEs was incubated with 1 nM biotin-labeled target DNA for 50 min on ice in the reaction buffer containing 20 mM Tris-HCl (pH 8.0), 150 mM NaCl, 5 mM $MgCl_2$, 10% glycerol, 10 ng per µl poly (dI·dC) and 0.1 mg per ml BSA. Reactions were then resolved on 8% native acrylamide gels in 0.5× TBE buffer at 15 V per cm for 1 h. Dried gels were exposed to Bio-Rad chemiluminescence and analyzed by Image Lab 3.0. The sequences of oligonucleotides used in EMSA experiments are shown below. The three tandem C (underlined) of the TALE-binding sequence (within square brackets) are chemical synthesized to C, 5mC, 5hmC or T.

MAPK6-Sense: 5′- TTCAGCTGGAT[CCCGGAGGA] GCGGATATAACCAGG -3′

**Statistical information.** All statistical results are expressed as mean ± s.d. The data of triplicate samples were used for statistical analyses. The significance between groups were evaluated by two-tailed unpaired Student's t test. *P < 0.05, and **P < 0.005

**Data availability.** The data that support the findings of this study are accompanied with the article and Supplementary files, or are available from the corresponding authors.

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

## Acknowledgements

We thank Zhonglin Fu for technical assistance. This work was supported by grants from the National Basic Research Program of China (MOST2016YFC0900301) to C.Y. and the National Natural Science Foundation of China (21522201 and 91519325 to C.Y., 31430025 and 81471909 to W.W.). This work was also supported by Beijing Advanced Innovation Center for Genomics at Peking University (to W.W.), and the Peking-Tsinghua Center for Life Sciences (to W.W. and C.Y.).

## Author contributions

Y.Z., L.L., S.G., J.S., C.Z., W.W. and C.Y. designed the research; Y.Z., L.L., S.G., J.S., C.Z. and Z.Y. performed the research; Y.Z., L.L., S.G., J.S., C.Z., W.W. and C.Y. analyzed the data; and Y.Z., S.G., J.S., C.Z., W.W. and C.Y. wrote the paper.
