## [Peer Review File · Nature Communications]

Reviewers' Comments:

Reviewer #1:

Remarks to the Author:

The manuscript entitled 'Complete decoding of TLA effectors for 5-methylcytosine and 5-hydroxymethylcytosine recognition' by Zhang et al. describes a systematic screening of 420 different RVD combinations, located at 3 consecutive repeats of an otherwise constant and canonical TAL effector DNA-binding region, for ability to tolerate and/or specifically recognize 5-methylcytosine (5mC) and/or 5-hydroxymethylcytosine (5hmC) at individual bases in a DNA target site. The rationale and motivation behind this work is to generate gene recognition platforms that are sensitive to positions and patterns of cytosine modification, in order to drive gene activation, repression, or modification in a manner that is responsive to the epigenetic status of individual bases.

The study consists of 4 components: First, an in cellulo screen for TAL effector binding activity, conducted with the panel of RVD variant-containing TAL effectors, using a GFP readout that is dependent on the ability each variant TAL effector to bind an upstream site containing C, 5MeC or 5hMeC. Second, a DNA binding competition assay to assess whether enhanced gene activation activity by certain TAL effector/DNA target combinations is correlated with enhanced DNA binding affinity for the same combination. Third and fourth, demonstrations that the RVD combinations identified in the screen could then be applied to create gene targeting platforms, for targeted gene activation or modification, that are sensitive to cytosine methylation status in living cells.

Overall, the study is a satisfying and effective examination of TAL effector action that exceeds most if not all prior published literature on TAL effector / RVD specificities. The inclusion of "X*" RVD combinations (constructs 401 to 420 in this study) and the comparative examination of C vs. 5-MeC vs. 5-hMeC go beyond prior studies of expansions of the RVD 'code', particularly the Miller et al. paper published in 2015 (reference 14). As well, study is a clear and important expansion of prior studies by Kubik et al. (references 31 and 32) in which a single discriminatory RVD ('HD', which recognizes C) versus a non-discriminatory RVD ('N*', which tolerates C and 5-mC) were used in a pairwise fashion to accomplish single-basepair resolution mapping of cytosine methylation status.

After reading through the entire manuscript and its associated materials, the only major query that I had written down was whether the relative activity of various TAL effector variants in the in cellulo gene activation assay (figures 1 and 2) truly correlates in a robust manner with the underlying sequence-specific DNA binding affinity of the same TAL effector variants (as stated in lines 110, 120, and 129) in the paragraph headed "Specific and degenerate RVDs for efficient 5mC and 5hmC recognition"). The authors back up that assertion with a subsequent analysis of relative DNA binding affinity that is conducted using a competition assay against a canonical restriction endonuclease. That assay provides an estimate of relative K_i values for pairs of TAL effectors, but doesn't really offer explicit binding affinities. It would be nice to see an orthogonal series of binding assays (via gel shifts or fluorescence polarization for example) but in the absence of that type of data, I think that the existing binding data illustrated in figure 3) is satisfactory.

Minor suggestions and issues:

Figure 2: Panel a shows the relative performance of each of 420 RVDs in discriminating between T, C, 5mC and 5hmC, as individual 4-panel heat plots for each RVD, graphed against a grid of 1st and 2nd RVD position. This is fine, but I would strongly recommend that they add a side 'legend' that blows up one four panel display for one RVD, and explicitly label which base corresponds to which position (upper left, upper right, lower left and lower right). That will be helpful to anyone who struggles with red versus green (such as myself) and also will help anyone who might be working with a photocopy or grey scale print-out of the figure.

Figure 2: Panel b shows the relative performance of a selected set of RVDs against C, 5mC and 54hmC. I would recommend adding the bars corresponding to the canonical 'HD' RVD to this figure so that the scale of signal and activity for the 'wild-type' or 'canonical' RVD against the variants in the figure can be visualized easily. Also, bright yellow against white is also hard to see--I recommend shading those bars just a bit more darkly.

Abstract: I generally discourage the use of 'For the first time' in manuscript and would recommend deleting it here.

Introduction:

Line 44: 'Utilizing the' would be better as 'Exploiting their'

Line 45: 'like' would be better as "such as'

Line 46: 'to become' would be better as 'to create'

Line 64: 'to recognize' would be more accurate if phrased 'to tolerate'.

Line 70: 'These observations hit' would be better as 'These observations illustrate'

Line 76: 'Change "Not only we find' to 'Not only do we find'

Line 76: 'regular C' would be better as 'unmodified C'.

Results:

Line 87: The authors state 'we utilize the screening platform that we developed previously'. This implies that the platform and method has been previously described. If so, then please provide a reference. If not, then I suggested changing to 'we utilize a screening platform that is described here'.

Line 152: "and at the meanwhile demonstrates' is awkward. I suggest changing to "and also demonstrates".

Line 167: 'demonstrates' should be 'demonstrated'.

Line 170: 'In full consistence with' would be better as 'Consistent with'.

Line 180: I recommend deleting 'Satisfactorily,' and simply stating 'We again observed...'.

Reviewer #2:

Remarks to the Author:

Transcription activator-like effector (TALE) proteins can recognize specific DNA sequences via their tandem repeat-variable diresidues (RVDs). TALE-based systems are powerful programmable gene-editing tools that can be fused with functional domains, like transcription activator, repressors or nucleotide endonucleases. In addition to target canonical unmodified DNA sequences, TALE proteins have been shown to recognize epigenetic DNA modifications, such as 5-methylcytosine (5mC) and 5-hydroxymethylcytosine (5hmC).

Built on their previously developed TALE RVD screening platform (Ref 13), authors systematically screened all possible TALE RVDs for 5mC and 5hmC in this manuscript. Authors reported three classes of RVDs: 1) universal (recognizing 5mC, 5hmC and regular C) and degenerate (recognizing both 5mC and 5hmC) RVDs; 2) "5mC-specific" RVDs (such as HA); 3) "5hmC-specific" RVDs (such as FS). Most of these RVDs are novel. These novel RVD-DNA recognitions are further validated by an in vitro protection assay. Importantly, authors also reported in vivo evidences of utilization of these novel RVDs to achieve methylation-dependent gene activation and genome editing. Finally, authors also report an example of locus-specific 5hmC detection in genomic DNA using the "5hmC-specific" RVD.

This is a comprehensive investigation of TALE recognition for 5mC and 5hmC and a proof-of-principal study of potential applications of these modified TALE proteins in epigenetic manipulation. Their work greatly expands our current knowledge of DNA recognition by TAL effectors. Overall, this manuscript deserves great attention in the fields of epigenetics and genome editing, however, there are a few issues that I feel need to be addressed before certain claims can be made. Therefore I have a few suggestions that I hope the authors can address:

1. Figure 2 and S2, authors reported that the identified RVDs also bind to regular T. Some RVDs (such as K*, H*, R*, G*, NG, and KG) have a higher binding preference for T than other modified cytosines. As author stated it is not surprising giving that they have either amino acid residues with small side chains or deletion of this residue at the 13th position (page 5, line 120-123). However, "5hmC-specific" RVD FS also has a higher binding preference for T than 5hmC (as shown in Figure S2). Therefore, in order to make a rigorous claim of RVDs are indeed 5hmC-specific or 5mC-specific RVDs (Figure 3), further in vitro validation assay is needed to include T as a control.

2. While it is not an issue for the up-regulation of EGFP in the screening system (the plasmid containing a pre-defined sequence), the preference of binding to regular T in addition to modified cytosine may raise concerns of potential off-target effect for other in vivo applications. It is conceivable that these TALE RVDs might target DNA sequences with identical flanking sequence but containing T instead of designed modified cytosines. Discussion of potential limitations or strategies for improving target specificity shall be included.

Minor point:

The authors' description of "Detection of 5hmC in the mammalian genome at single-base resolution" is inaccurate and misleading, as authors only target a single site in their study. It shall be rephrased as "Detection of locus-specific 5hmC in the mammalian genome".

Point-to-point response to reviewers' comments

Reviewer #1:

We thank this referee for the very positive comments and constructive suggestions! To address the specific comments:

Major comments:

1. “After reading through the entire manuscript and its associated materials, the only major query that I had written down was whether the relative activity of various TAL effector variants in the in cellulo gene activation assay (figures 1 and 2) truly correlates in a robust manner with the underlying sequence-specific DNA binding affinity of the same TAL effector variants (as stated in lines 110, 120, and 129) in the paragraph headed "Specific and degenerate RVDs for efficient 5mC and 5hmC recognition"). The authors back up that assertion with a subsequent analysis of relative DNA binding affinity that is

conducted using a competition assay against a canonical restriction endonuclease. That assay provides an estimate of relative K_i values for pairs of TAL effectors, but doesn't really offer explicit binding affinities. It would be nice to see an orthogonal series of binding assays (via gel shifts or fluorescence polarization for example) but in the absence of that type of data, I think that the existing binding data illustrated in figure 3) is satisfactory.”

Response: The reviewer is right that the *in vitro* protection assay does not provide explicit binding profiles. To explore the binding affinities of TALE RVDs to different cytosines *in vitro*, we have carried out an orthogonal assay (EMSA) during the revision, as suggested by the reviewer. In order to maintain consistency with our protection assay in Fig. 3, we used the same set of probes, which are also derived from the *MAPK6* gene and contain one modified cytosine in the middle of the sequence. As shown in the figure below, HD exhibits higher binding affinity to C comparing to 5mC, while RG preferentially recognize 5mC than unmodified C. These results are consistent with our observations from our *in cellulo* (Fig.1 and 2) and *in vitro* (Fig.3) assays.

Binding affinity of HD and RG to DNA probes containing one single cytosine modification by EMSA (reviewer-only figure)

We also searched literature for EMSA experiments using TALE proteins, and found that previous studies have used multiple modified bases (5~6 modification sites in one DNA probe) for the binding assay^{1,2}. We thus tried to examine whether the binding affinity can be influenced by the increased number of modified cytosines. We chemically synthesized additional DNA probes, which contain three 5mC, 5hmC or unmodified C at the defined positions (the sequences are listed in the revised Supplementary methods). To recognize the modification status of these defined positions, we have to re-assemble new TALE constructs and purify the corresponding TALE proteins with varying RVDs. We then carried out EMSA to profile their binding abilities. The results (see the newly added Supplementary Figures 3d and 3e) again agree very well with our *in cellulo* and *in vitro* observations: first, HD binds strongly to C but poorly to 5mC and 5hmC; second, HA specifically recognizes 5mC but not other cytosines; third, RG binds well to both 5mC and 5hmC but

not C; fourth, FS preferentially recognizes 5hmC than 5mC and C. During our EMSA validation, we also found that the exact K_d values are dependent on the conditions (length of DNA, pH, salt and competing DNA concentration) of the EMSA experiments; for instance, the concentration of competing DNA can affect the measured K_d values (see the figure below). Hence, we think that although the absolute K_d values derived from the EMSA experiments can vary, the relative binding abilities provide useful comparisons among different cytosines. Indeed, the relative affinities and the trend of novel RVDs in recognizing cytosine modifications obtained from the orthogonal EMSA experiments are very consistent with our *in cellulo* and *in vitro* observations.

The concentration of competing DNA can affect the measured K_d values for EMSA. For example, the measured K_d values of HA are much smaller when 5ng/ μ l competing DNA (figure a) is added than 10ng/ μ l competing DNA (figure b).(reviewer-only figure)

In addition to these new experiments, we have also modified the corresponding text to clearly separate the results of activation efficiency and that of binding affinity. For instance, we have changed the first sentence in this paragraph (line 109-110) to “We identified multiple 5mC binders that showed high activation efficiency in the above *in cellulo* assay and group them into three categories...”. We also changed the sentence in line 120 to “... demonstrate higher activation efficiencies for 5mC-containing reporters.” And in 129 to “Although their abilities in activating 5hmC-containing reporters are lower than the universal binders, ...”.

Minor comments

1. “Figure 2: Panel a shows the relative performance of each of 420 RVDs in discriminating between T, C, 5mC and 5hmC, as individual 4-panel heat plots for each RVD, graphed against a grid of 1st and 2nd RVD position. This is fine, but I would strongly recommend that they add a side 'legend' that blows up one four panel display for one RVD, and explicitly label which base corresponds to which position (upper left, upper right, lower left and lower right). That will be helpful to anyone who struggles with red versus green (such as myself) and also will help anyone who might be working with a photocopy or grey scale print-out of the figure.”

Response: Thanks for your kind advice! We have added a side legend in the upper right corner of Panel a, Figure 2, which blows up the specific heat map of Y*. To make it clear, we also added the statement in the legend of figure 2, as “The upper right side legend shows the specific heat map of Y*. Starting from top left, the DNA reporter is 5mC, 5hmC, T and C, respectively”.

2. “Figure 2: Panel b shows the relative performance of a selected set of RVDs against C, 5mC and 5hmC. I would recommend adding the bars corresponding to the canonical 'HD' RVD to this figure so that the scale of signal and activity for the 'wild-type' or 'canonical' RVD against the variants in the figure can be visualized easily. Also, bright yellow against white is also hard to see--I recommend shading those bars just a bit more darkly.”

Response: We agree with the referee and have added the bars corresponding to HD to the revised figure 2. We have also changed the 5hmC bar to a darker color to allow easier visualization.

1. “Abstract: I generally discourage the use of 'For the first time' in manuscript and would recommend deleting it here.”

Response: We have deleted the statement of “For the first time” in Abstract as suggested.

2. “Line 44: 'Utilizing the' would be better as 'Exploiting their'”

Response: We have revised the statement to “Exploiting their modular DNA-recognition property, ...”.

3. “Line 45: 'like' would be better as "such as"”

Response: We have changed “like” to “such as”.

4. “Line 46: 'to become' would be better as 'to create'”

Response: We have changed “to become” to “to create”.

5. “Line 64: 'to recognize' would be more accurate if phrased 'to tolerate'.”

Response: We have changed “to recognize” to “to tolerate”.

6. “Line 70: 'These observations hit' would be better as 'These observations illustrate'”

Response: We have changed “hint” to “illustrate”.

7. “Line 76: 'Change "Not only we find' to 'Not only do we find'”

Response: We have revised the sentence to “Not only do we find”.

8. “Line 76: 'regular C' would be better as 'unmodified C'.”

Response: We have made the recommended revision. To make the description consistent throughout the entire manuscript, we also made additional revisions in Line 114, Line 121, Line 136, Line 146, Line 172, and Line 482, respectively.

9. “Line 87: The authors state 'we utilize the screening platform that we developed previously'. This implies that the platform and method has been previously described. If so, then please provide a reference. If not, then I suggested changing to 'we utilize a screening platform that is described here'.”

Response: The platform we utilized in this study has been elaborated in our previous study³. We have mentioned this key paper in the introduction part but we forgot to cite this reference in this particular statement in our last version of manuscript. In the revised manuscript, we have added this citation in Line 87.

10. “Line 152: "and at the meanwhile demonstrates' is awkward. I suggest changing to "and also demonstrates".”

Response: We have substituted “and at the meanwhile” to “and also”.

11. “Line 167: 'demonstrates' should be 'demonstrated'.”

Response: We have changed “demonstrates” to “demonstrated”.

12. “Line 170: 'In full consistence with' would be better as 'Consistent with'.”

Response: We have revised “In full consistence with” to “Consistent with”.

13. “Line 180: I recommend deleting 'Satisfactorily,' and simply stating 'We again observed...'. ”

Response: We have deleted “Satisfactorily” in the revised manuscript. We want to thank the referee again for the very careful and constructive review!

Reviewer #2:

We thank this referee for the very positive comments and constructive suggestions! To address the specific comments:

Major comments:

1. “Figure 2 and S2, authors reported that the identified RVDs also bind to regular T. Some RVDs (such as K*, H*, R*, G*, NG, and KG) have a higher binding preference for T than other modified cytosines. As author stated it is not surprising giving that they have either amino acid residues with small side chains or deletion of this residue at the 13th position (page 5, line 120-123). However, “5hmC-specific” RVD FS also has a higher binding preference for T than 5hmC (as shown in Figure S2). Therefore, in order to make a rigorous claim of RVDs are indeed 5hmC-specific or 5mC-specific RVDs (Figure 3), further *in vitro* validation assay is needed to include T as a control.”

Response: It is true that the identified RVDs that recognize the modified cytosines also bind to regular T in our *in cellulo* experiments (Figs. 1 and 2). To further compare the binding affinity of RVDs to modified cytosines and regular T, we carried out additional *in vitro* validation assay as the reviewer suggested. Because the *MspI* restriction enzyme recognizes the -CCGG- restriction site, we could not use the protection assay to directly compare between C and T. Therefore, we utilized electrophoretic mobility shift assay (EMSA) to compare the binding affinity of RVDs to modified cytosines and T. To do so, we chemically synthesized new DNA probes containing either modified cytosines, unmodified C or T. We also re-assembled new TALE constructs and purified the corresponding TALE proteins to carry out the gel shift assays (the newly added Supplementary Fig. 3d and 3e). RG is a degenerative binder for modified C and can also accommodate T in our *in cellulo* experiments; indeed it binds to T as well in the EMSA results. The similar situation is also observed for FS, which recognizes both 5hmC and T. For HA, its binding affinity is slightly better for 5mC than T in the *in cellulo* experiments; interestingly, it appears to very specific for 5mC in the EMSA experiments (the second column in the newly added Supplementary Fig. 3d and 3e). Nevertheless, these results suggest that some of the binders for modified cytosines could indeed bind regular T. To avoid potential misinterpretation, we have changed the heading of this paragraph into “Specific and degenerate RVDs for efficient 5mC and 5hmC recognition among cytosines”. We have also added several sentences in the Discussion part to point out these observations and potential limitations.

2. “While it is not an issue for the up-regulation of EGFP in the screening system (the plasmid containing a pre-defined sequence), the preference of

binding to regular T in addition to modified cytosine may raise concerns of potential off-target effect for other in vivo applications. It is conceivable that these TALE RVDs might target DNA sequences with identical flanking sequence but containing T instead of designed modified cytosines. Discussion of potential limitations or strategies for improving target specificity shall be included. ”

Response: We agree with the referee. We have added several sentences in the revised Discussion part (NG is the natural binder of T; some of the specific (FS) and degenerate (RG) RVDs we identified, which has small amino acids at the 13th position, could also accommodate T (Supplementary Figs. 3d and 3e). Hence, cautions should be taken that DNA sequences with identical flanking sequence but containing either modified cytosines or T could both be targeted during potential in vivo applications.) to discuss the potential limitations.

Minor comments:

“The authors’ description of “Detection of 5hmC in the mammalian genome at single-base resolution” is inaccurate and misleading, as authors only target a single site in their study. It shall be rephrased as “Detection of locus-specific 5hmC in the mammalian genome”.”

Response: We agree. We have changed the subtitle of Paragraph 2 in Page 9 to “RVD-mediated detection of locus-specific 5hmC in the mammalian genome” as suggested.

Reference

1. Deng, D. et al. Recognition of methylated DNA by TAL effectors. *Cell Res* **22**, 1502-1504 (2012).
2. Kubik, G., Schmidt, M.J., Penner, J.E. & Summerer, D. Programmable and highly resolved in vitro detection of 5-methylcytosine by TALEs. *Angew Chem Int Ed Engl* **53**, 6002-6006 (2014).
3. Yang, J. et al. Complete decoding of TAL effectors for DNA recognition. *Cell Res* **24**, 628-631 (2014).

Reviewers' Comments:

Reviewer #1:

None

Reviewer #2:

Remarks to the Author:

The authors have addressed all of my concerns and comments with new supporting data and revision in this revised manuscript. Therefore, in this reviewer's opinion, the revised manuscript is ready for acceptance for publication.